# Analysis and Evaluation of the Progressive Collapse Behaviour of a Cable Dome Structure

**Lianmeng Chen [1,*], Zebin Li [1], Yijie Liu [1], Kaiyu Huang [1], Yihong Zeng [1], Yiyi Zhou [2] and Shilin Dong [3]**

[1]   College of Civil Engineering and Architecture, Wenzhou University, Wenzhou 325035, China
[2]   College of Civil Engineering and Architecture, Changzhou Institute of Technology, Changzhou 213002, China
[3]   Space Structures Research Center, Zhejiang University, Hangzhou 310027, China
*   Correspondence: 00151034@wzu.edu.cn; Tel.: +86-13957790090

**Abstract:** In this study, the progressive collapse behaviour of a cable dome structure was analysed and evaluated according to the importance of element. First, the dynamic response and collapse mode caused by the removal of different types of cables and struts from a cable dome structure were studied using the instantaneous unloading method of full dynamic equivalent load. Second, a method was developed for element importance classification based on collapse modes, and the importance coefficient was introduced after comparing the node displacements before and after the removal of different elements. On this basis, the correlations of the importance coefficient of an element with its importance classification and the collapse mode caused by its removal were examined. Third, the influences of some design parameters on the resistance of cable dome structures to progressive collapses and on the importance coefficients of components were analysed and evaluated. Finally, a method was proposed to determine the critical value of the element importance category. The results of this study indicated that Cable-Strut elements differed in their antiprogressive collapse effects and importance coefficients, and thus produced different dynamic responses and collapse modes when they were removed. Cable domes differed in their critical importance coefficients for Cable-Strut elements, and design parameters differed in their influence on the antiprogressive collapse resistance of cable domes.

**Keywords:** cable dome structure; progressive collapse; dynamic response; element importance analysis; parameter analysis

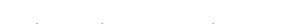



## 1. Introduction

Cable dome structures are a type of flexible tension structure based on Fuller's idea of tensegrity [1]. They have many advantages, such as high bearing performance, crossing capacity and a lightweight nature. Cable dome structures are widely used in engineering applications, and the commonly used types are the Geiger cable dome [2] and Levy cable dome [3]. However, because of the low redundancy of these structures, they easily undergo progressive collapse when subjected to overload, explosion and other unexpected conditions; that is, the chain reaction caused by the initial partial damage eventually leads to the overall or large-scale collapse of the structure.

The earlier studies on the progressive collapse of building structures have mostly focused on frame structures [4,5], and few studies have been conducted on long-span structures. Conventional long-span structures, such as grid frame (or shell) structures, are in general highly statically indeterminate structures, and the failure of a single component does not considerably reduce the bearing capacity of these structures. However, tension structures, such as cable dome structures, differ from conventional long-span structures with high-order statically indeterminate components. They have low redundancy, are sensitive to unexpected disturbances and are prone to collapse events under overload or unexpected disturbances [6,7]. Therefore, the analysis of the progressive collapse of cable

dome structures has considerable theoretical and practical value, and some research has been gradually conducted on the progressive collapse failure of these structures in the last decade.

Yuan et al. used the life-and-death unit technology to analyse the cable breaking of Levy cable dome structures and found that the structural response after partial Cable-Strut element failure could be used to judge whether the structure would undergo progressive collapse [8]. Zhao et al. analysed the cable breakage of the cable-membrane structure of Bao'an Stadium (in Shenzhen, China) and found that the breakage of a single radial cable would cause high local deformation. Moreover, the breakage of a single hoop cable caused high deformation and the relaxation of the entire structure, which would lead to the failure of the whole structure [9]. Lu et al. used an explicit dynamic integration method to simulate the breaking of a local cable in a full-tension, self-balancing Cable-Strut structure. Their results revealed its superior ability to resist progressive collapse [10]. Chen et al. analysed the mechanisms of the cable relaxation and cable failure exit of Kiewitt-type cable dome structures under various loading conditions. They found that the failure of a cable usually led to the redistribution of the internal forces of Cable-Strut elements in local areas without the failure of the entire structure, whereas the failure of a hoop cable led to the overturning of the entire structural system [11]. Tang et al. adopted the instantaneous loading method to simulate the failure of different types of components in a tensioned cable-membrane structure and analysed the structural responses after these components failure by considering material and geometric nonlinearities. They found that the back cables on both sides were key components that affected the overall performance of the aforementioned structure [12]. Zhang et al. studied a Geiger cable dome structure using the demolishing component method and found that the collapse resistance of this structure can be increased by preventing the failure of its outer hoop cable [13].

Fan et al. conducted research on the large grid-inclined, chord-supported dome structure in Dalian Gymnasium (Liaoning Province, China) and found that the breakage of a radial cable was more likely to cause the local collapse of this structure than was the breaking of a hoop cable [14]. Jiang used LS-DYNA software to conduct a nonlinear dynamic analysis of long-span truss structures and typical tension-string beams and trusses. They also analysed the anticollapse performance of the planar system and the collapse resistance effect of out-of-plane auxiliary structural members [15]. Cai et al. analysed the progressive collapse of the cable-arch roof system of New Guangzhou Railway Station (in Guangdong Province, China) by shifting the load path. They found that the use of a new cable-arch structure and giant truss structure effectively enhanced the progressive collapse resistance of the aforementioned system [16]. Zeng et al. analysed the dynamic response and antiprogressive collapse of four structural designs with a high number of spare cables and found that the spare cables effectively improved the antiprogressive collapse capability of the structural when the horizontal stiffness of the support was low [17]. Qu et al. proposed that the safety reserve should be considered in the design of the chord-supported dome structure of Zhaoqing New Area Gymnasium (in Guangdong Province, China). They suggested that the safety levels of key parts of this structure should be improved [18]. The numerical and theoretical methods proposed by Xu et al. [19] and the mixed finite-element model proposed by Tian et al. [20] were suitable for investigating the progressive collapse behaviour and static and dynamic responses of dome structures. Shekastehband et al. presented experimental and numerical studies on the collapse behaviour of tensegrity systems considering cable rupture and strut collapse [21,22]. Kiakojouri et al. discussed the impact of some parameters, such as topology of the structure, nature of the triggering event, size of the initial failure and seismic design requirements, on the strengthening and retrofitting strategy, and then proposed a comprehensive review on strengthening and retrofitting techniques to mitigate progressive collapse [23].

Thus, considerable studies have been conducted to investigate the internal force, structural displacement and the collapse caused by the breakage of Cable-Strut elements or failure of partial elements; however, most of these studies have only conducted qualitative

examinations of cable relaxation and withdrawal from work, local large deformation and large displacement. Studies have not comprehensively investigated structural collapse mechanisms, the importance of various types of Cable-Strut elements in resisting the progressive collapse of structures and the influence of design parameters on the resistance of structures to progressive collapse.

Accordingly, in this study, the failure mode and collapse mechanism of a real stadium with a Geiger cable dome structure was investigated, and the importance of various components in resisting progressive collapse was quantitatively evaluated. First, using LS-DYNA software (Ansys), this study explored the internal force, displacement and energy response of the selected cable dome structure during progressive collapse caused by local component failure. Subsequently, failure mode identification and element importance analysis were performed for various components. Second, structural components were classified into different categories according to their importance coefficients, which were determined by comparing the changes in node displacement before and after the corresponding component removal. On this basis, the correlations of the importance coefficient of an element with its importance classification and the collapse mode caused by its removal were examined. Third, according to the collapse mode caused by cable (or strut) removal, a critical importance coefficient was selected to classify the properties of Cable-Strut elements. Finally, the influences of the design parameters of Cable-Strut structures on their resistance to progressive collapse, the importance of different elements and the critical importance coefficient of these structures were analysed.

## 2. Structural Model and Calculation Method

### 2.1. Cable Dome Structure of the Selected Sports Centre

The structure investigated in this study was a Geiger cable dome structure of the roof of a sports centre in Yi Qi, Inner Mongolia, China. The span of the cable dome is 71.2 m, the rise of the cable dome is 5.5 m, and the ratio of the rise of the cable dome to its span is approximately 1/13. The investigated cable dome structure comprises 20 symmetrical Cable-Strut units, and the design load of this cable dome is 0.4 kN/m$^2$. The dome has two hoop cables, and a tension hoop is present at the dome centre. The entire structure is fixed on and supported by the surrounding rigid compression circumferential beams. The structural model, structural plan and structural profile are displayed in Figure 1. The design parameters of different elements are displayed in Table 1. The elastic modulus values of the cables and struts are 160 and 206 GPa, respectively.

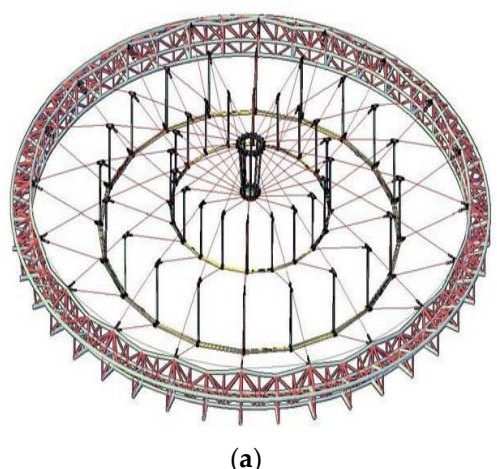

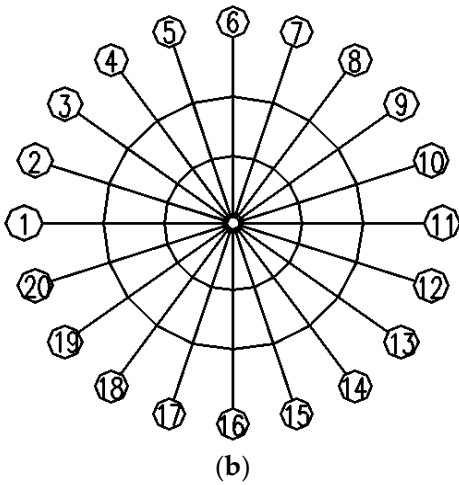

(**a**)  (**b**)

**Figure 1.** *Cont.*

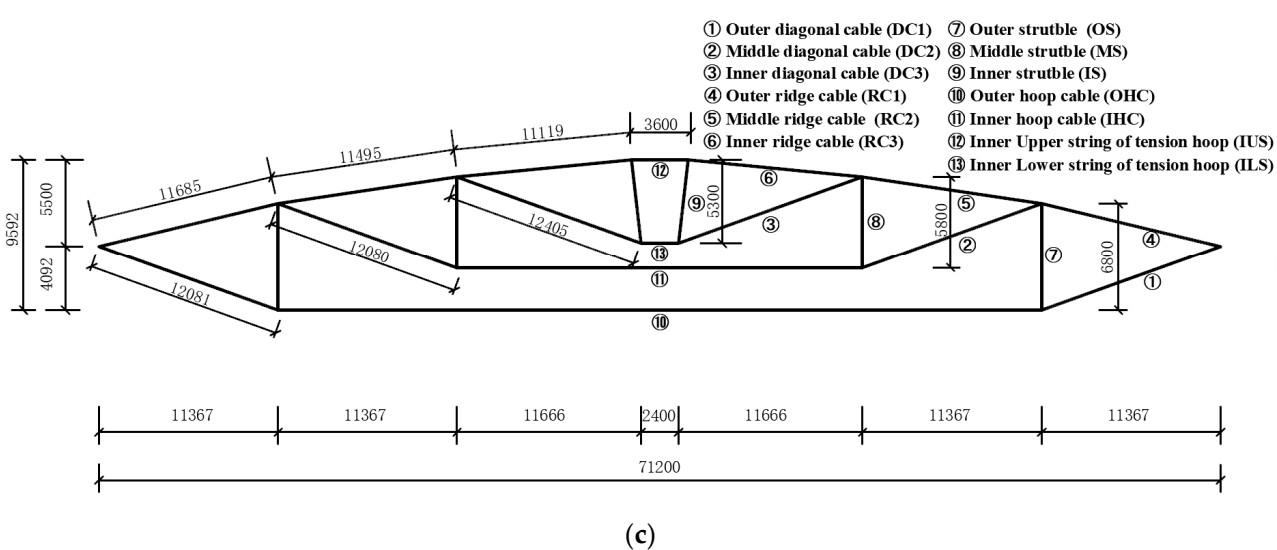

**(c)**

**Figure 1.** Investigated cable dome structure. (**a**) Structural model, (**b**) Structural plan, (**c**) Structural profile and size.

**Table 1.** Sectional area and initial prestress of different elements.

| Component | DC1 | DC2 | DC3 | RC1 | RC2 | RC3 | OS | MS | IS | OHC | IHC | IUS | ILS |
|---|---|---|---|---|---|---|---|---|---|---|---|---|---|
| Sectional area ($mm^2$) | 2490 | 853 | 605 | 1840 | 1360 | 853 | 7800 | 4670 | 4670 | 7470 | 3320 | 3320 | 3320 |
| Prestress (kN) | 466.6 | 208 | 105.9 | 682.2 | 473.1 | 370 | −158 | −70.4 | −36.2 | 1403.2 | 625.7 | 1190.1 | 305.3 |

*2.2. Analytical Methods*

2.2.1. Selection and Modelling of Cable-Strut Units

On the basis of the mechanical properties of the Cable-Strut elements of the investigated structure, LINK167 and LINK160 were selected as the cable and strut elements in LS-DYNA. The prestress was calculated using the following equation:

$$F = K \times max\{\Delta L, 0.0\} \tag{1}$$

$$K = EA/(L_0\text{-offset}) \tag{2}$$

where F represented the prestress; K was the structural coefficient of the structure; $\Delta L$ and $L_0$ were the variation in the cable length and the initial cable length (or strut), respectively; E and A were the elastic modulus and sectional area of the cable (or strut), respectively; and offset was the offset quantity. The bilinear dynamic material model was used to model the LINK160 element. The failure strain of a strut was defined as 0.01; that is, in the analysis process, if the strain of a compression strut exceeded 0.01, then the strut was automatically deleted from the structure.

2.2.2. Removal and Replacement of Cable-Strut Elements with Equal Force

To account for the initial state of the investigated structure and eliminate the dynamic influence of an increase in the static load on the structure, the instantaneous unloading method of full dynamic equivalent load was adopted in this study. Thus, a component was removed from the structure and replaced with equal force. This force was unloaded, which was equivalent to that caused by the complete removal the corresponding component, to analyse the dynamic response under component failure and then evaluate the importance of structural elements.

Structures are subject to the problem of natural vibration when one of their components is replaced by an equal force, which is then unloaded. In general, the replacement time, duration time and unloading time of equal force are 2, 20 and 1/10 times that of the natural

vibration period of the residual structure [24,25], respectively. Modal analysis results of the investigated cable dome structure indicated that the maximum natural vibration period of the residual structure was 1.6025 s. This figure was rounded to 2 s in this study for convenience in calculation. The replacement of the cable-strut with equal force was completed in a linear increase within 4 s. After another 40 s, the structure tended to be stable under the action of structural damping. The unloading of the equal force began from 44 s and was complete after a linear increase over less than 0.2 s. Subsequently, the internal force of the structure began to be redistributed, and the equilibrium state was reached within 200 s. The specific action time of the equal force is presented in Figure 2.

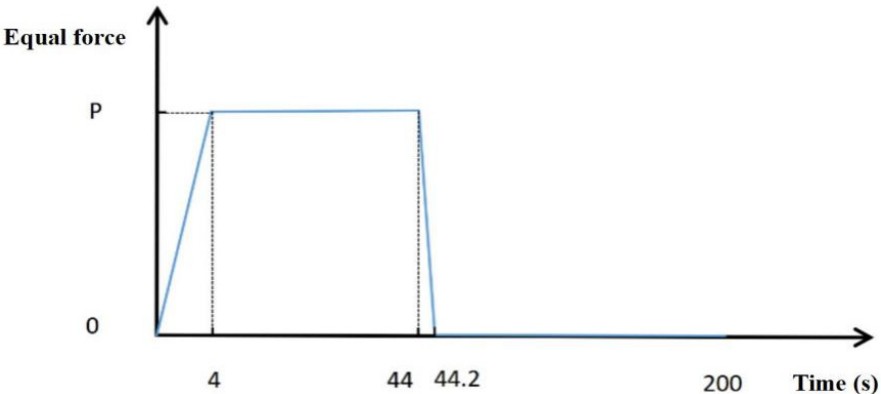

**Figure 2.** Equal force schedule.

## 3. Structural Dynamic Response after the Removal of an Element and Element Importance Analysis in the Progressive Collapse Mode

Because of space limitations, the inner hoop cable (IHC) was used as an example in the following section to determine the displacement, internal force, energy response and collapse mode of the investigated structure after the removal of an element.

### 3.1. Dynamic Response and Collapse Mode of the Investigated Structure

#### 3.1.1. Displacement Response Analysis

Figure 3a depicts the vertical displacement diagram for the situation in which the investigated structure began to unload the equal force (t = 44 s). Figure 3b displays the vertical displacement diagram for the situation in which the structure reached a new force equilibrium state (t = 200 s) under damping. These figures indicate the following. First, after the IHC was removed, nodes 4 and 11, which were connected to the IHC, moved quickly to both sides of the hoop; thus, all the other IHC elements moved horizontally to varying extents, which caused all the Cable-Strut elements connected with the IHC to have varying horizontal displacement. Second, because of the failure of the IHC and the weakening of the support provided by it to the middle strut (MS), all the upper and lower nodes of the MS had large vertical displacements. The upper and lower nodes of the MS that were directly connected to the failed strut exhibited considerable changes in displacement. Nodes 4 and 5 had vertical displacements of 0.531 and 2.87 m, respectively. Third, after the displacement of the aforementioned nodes, the upper and lower nodes of the MS in adjacent trusses were displaced. The displacement of the nodes in the other trusses was negatively correlated with the distance from the failed Cable-Strut element.

#### 3.1.2. Energy Response Analysis

Figure 4 illustrates the kinetic energy diagram over a period of 200 s (t) and indicates the following. First, when an equal force replaced a Cable-Strut element, the kinetic energy response was high initially and then decreased gradually to 0 within 44 s under the effect of structural damping. Second, the dome structure exhibited a higher kinetic energy response than that observed in the equilibrium state, and the influence range was larger when the equivalent load was removed at t = 44 s.

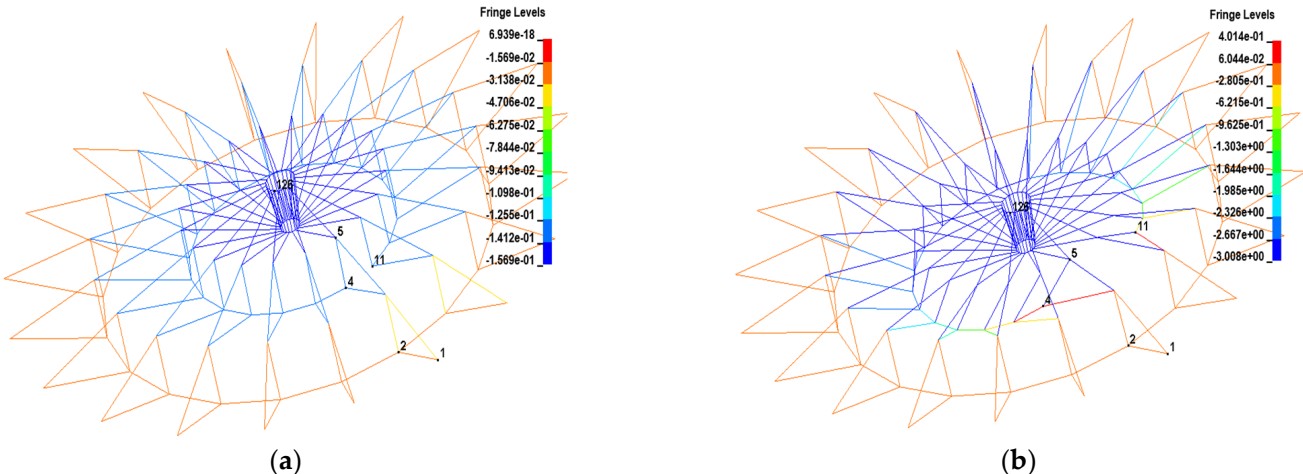

(**a**)                                    (**b**)

**Figure 3.** Variations in the vertical displacement over time. (**a**) t = 44 s, (**b**) t = 200 s.

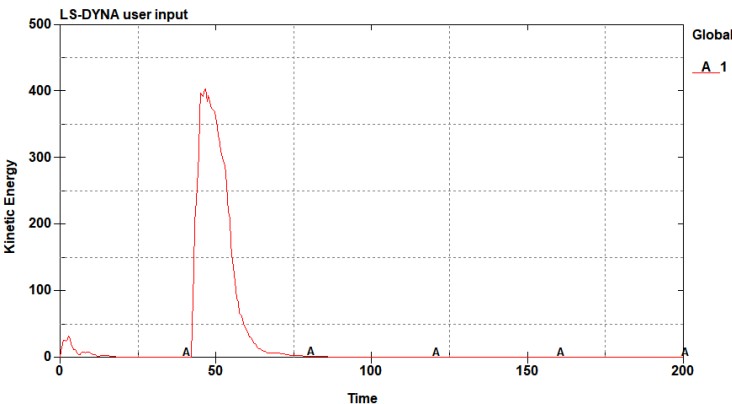

**Figure 4.** Variations in the kinetic energy with time.

### 3.1.3. Internal Force Response Analysis

Figure 5 shows the variations in the internal force of the left adjacent truss of the failed Cable-Strut element with time. As displayed in this figure, the internal forces of the adjacent cables became 0, and the failure occurred when the IHC was removed. In addition, the internal forces of the other cables and struts exhibited a high prestress loss. The closer a strut was to failure, the higher was its prestress loss. The internal force of the inner diagonal cable (DC3) decreased the least by 29%.

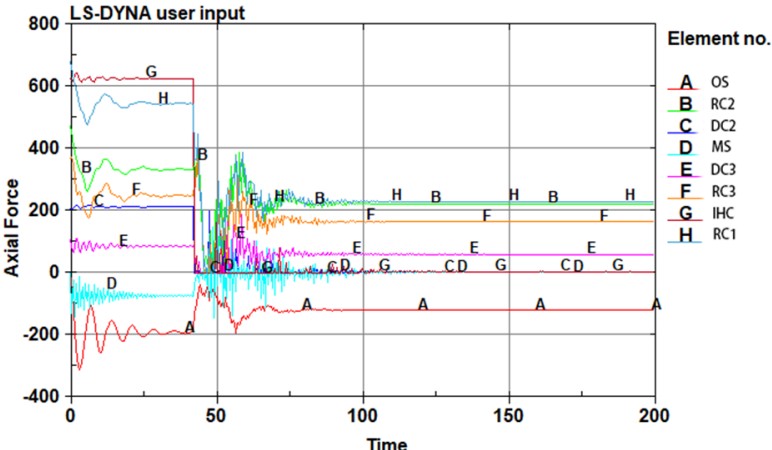

**Figure 5.** Variations in the internal force with time.

### 3.1.4. Collapse Mode Analysis

The overall internal force of the elements surrounding the outer hoop cable (OHC) increased, and the displacement of the components inside the OHC was larger than 1/50 times the span when the IHC was removed. When the investigated dome structure reached the final equilibrium, the collapse area reached 47%, and the collapse scope belonged to complete destruction. The maximum vertical displacement of 3 m occurred at node 126 (the upper node of the internal strut).

### 3.2. Importance Analysis of Cable-Strut Elements According to Collapse Mode

Table 2 presents the dynamic response and collapse mode of the investigated structure after removing each Cable-Strut element sequentially. Different types of dynamic responses and collapse modes were caused by the removal of different types of Cable-Strut elements. The removals of the IHC and OHC resulted in the formation of a large collapse area and the occurrence of a large vertical displacement. A moderately large collapse deformation area was formed when the upper string of the tension hoop was removed. Moreover, a relatively small collapse deformation area was formed when other elements, such as the outer diagonal cable, middle diagonal cable, inner diagonal cable, outer ridge cable middle ridge cable, inner ridge cable, outer strut (OS), MS, inner strut and lower string of the tension hoop, were removed. Thus, different types of Cable-Strut elements differed in how important they were to the stability of cable domes.

**Table 2.** Types of collapse induced by the removal of different types of Cable-Strut elements and the importance categories of these elements.

| Cable-Strut | Description of Collapse | Collapse Models | Importance Coefficient | Important Properties |
|---|---|---|---|---|
| Outer diagonal cable | 10% of the collapsed area; the maximum vertical displacement was 0.64 m at Node 3 (upper node of the outer strut) | Non-progressive collapse | 0.01 | Common component |
| Middle diagonal cable | 0% of the collapsed area; the maximum vertical displacement was 0.36 m at Node 5 (upper node of the middle strut) | Non-progressive collapse | 0.0049 | Common component |
| Inner diagonal cable | 0% of the collapsed area; the maximum vertical displacement was 0.10 m at Node 7 (upper node of the inner strut) | Non-progressive collapse | 0.0009 | Common component |
| Outer ridge cable | 10% of the collapsed area; the maximum vertical displacement was 9.93 m at Node 3 (upper node of the outer strut) | Non-progressive collapse | 0.023 | Common component |
| Middle ridge cable | 4.6% of the collapsed area; the maximum vertical displacement was 8.30 m at Node 5 (upper node of the middle strut) | Non-progressive collapse | 0.01 | Common component |
| Inner ridge cable | 0% of the collapsed area; the maximum vertical displacement was 0.10 m at Node 7 (upper node of the inner strut) | Non-progressive collapse | 0.0015 | Common component |
| Outer strut | 0% of the collapsed area; the maximum vertical displacement was 1.42 m at Node 3 (upper node of the outer strut) | Non-progressive collapse | 0.0061 | Common component |
| Middle strut | 0% of the collapsed area; the maximum vertical displacement was 1.10 m at Node 5 (upper node of the middle strut) | Non-progressive collapse | 0.002 | Common component |
| Inner strut | 0% of the collapsed area; the maximum vertical displacement was 0.10 m at Node 7 (upper node of the inner strut) | Non-progressive collapse | 0.000074 | Common component |

**Table 2.** *Cont.*

| Cable-Strut | Description of Collapse | Collapse Models | Importance Coefficient | Important Properties |
|---|---|---|---|---|
| Outer hoop cable | 100% of the collapsed area; the maximum vertical displacement was 5.58 m at Node 136 (upper node of the outer strut) | progressive collapse | 0.54 | Key component |
| Inner hoop cable | 47% of the collapsed area; the maximum vertical displacement was 3 m at Node 126 (upper node of the inner strut) | progressive collapse | 0.23 | Key component |
| Inner upper string of tension hoop | 16.6% of the collapsed area; the maximum vertical displacement was 1.56 m at Node 3 (upper node of the outer strut) | Partial progressive collapse | 0.15 | Important component |
| Inner lower string of tension hoop | 16.6% of the collapsed area; the maximum vertical displacement was 0.756 m at Node 7 (upper node of the inner strut) | Non-progressive collapse | 0.033 | Common component |

According to the UFC4-023-03 standard in the United States [24], the collapse of cable dome structures has one of three modes: progressive collapse, partial progressive collapse and nonprogressive collapse. Progressive collapse occurs when the maximum vertical node displacement of a cable dome structure is larger than 1/50 times its span and the failure area covers 30% of its total plane area. Partial collapse occurs when the maximum vertical node displacement of a cable dome structure is larger than 1/50 times its span, but the failure area is less than 30% of its total plane area. Nonprogressive collapse occurs when the maximum vertical node displacement of a cable dome structure is smaller than 1/50 times its span or when the maximum node displacement of the cable dome is larger than 1/50 times its span, but the failure area is less than 15% of its total plane area.

According to the collapse mode caused by the removal of a Cable-Strut element, the element was categorised as a key component, an important component or a common component (Table 2).

### 3.3. Definition of Importance Coefficient of a Cable-Strut Element

In order to describe quantitatively the importance of each element, the importance coefficient of a Cable-Strut element was proposed and expressed as follows:

$$\rho_i = \frac{\|s_{1i} - s_0\|}{\|s_0\|} \tag{3}$$

where $\| \ \|$ was the European norm and $s_0$ and $s_{1i}$ were the displacement vectors of the node displacement of the cable dome structure under the same load before and after the removal of component $i$, respectively. The parameter $\rho_i$ was proportional to the displacement caused by the removal of component $i$. The higher the value of $\rho_i$, the more important the corresponding Cable-Strut element to the antiprogressive collapse of the structure, and the stronger the effect.

Each Cable-Strut element was removed separately for the calculation of the importance coefficient of the corresponding element. The normalisation operation expressed in Equation (4) was used to perform a unified quantitative comparison of all coefficients, and the corresponding results are shown in Figure 6.

$$\beta_j = \rho_j / \sum_{j=1}^{13} \rho_j \tag{4}$$

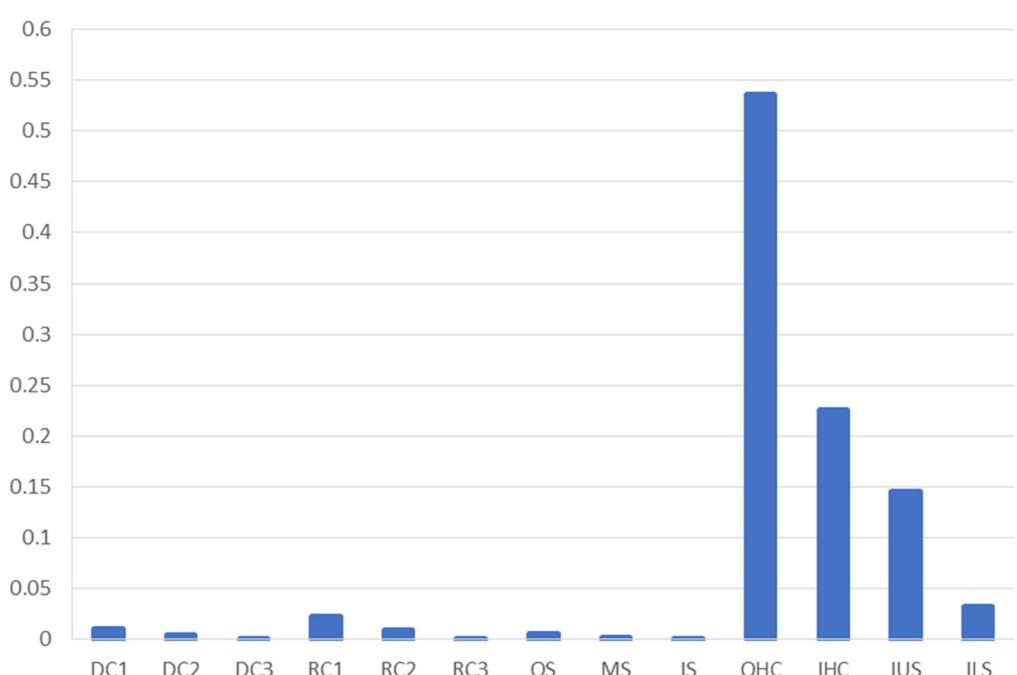

**Figure 6.** Importance coefficients of each type of Cable-Strut element.

The OHC and IHC had the highest and second-highest element importance coefficients, respectively. After the removal of these cables, the investigated structure exhibited a strong dynamic response, which led to progressive collapse; thus, the OHC and IHC were key components of the investigated structure. The removal of the inner upper string (IUS) of the tension hoop caused the local progressive collapse of the investigated structure; thus, this component moderately affected the stability of the aforementioned structure and thus had a high importance coefficient. The removal of the other types of Cable-Strut elements had a marginal effect on the structure and did not cause progressive collapse. In general, the order of importance coefficients of the different types of Cable-Strut elements was hoop cables > ridge cables > diagonal cables > struts.

## 4. Influences of Design Parameters on the Structural Resistance to Progressive Collapse and on the Importance Coefficients of Components

To explore the influences of different design parameters on the resistance of cable dome structures to progressive collapse and on the importance coefficients of Cable-Strut elements, the dynamic response and collapse mode were investigated under different structural parameters, such as the cross-sectional area of a component, the radius of a hoop cable, the length of a strut, the number of structural trusses and the structural topological relationship.

### 4.1. Influence of the Cross-Sectional Area of a Cable-Strut Element on the Collapse Resistance

The cross-sectional area of a Cable-Strut element was set as 0.5, 0.75, 1.25 and 1.5 times the initial cross-sectional area, respectively, and the influences of the cross-sectional area of each element on the importance coefficients of components were showed in Figure 7. The cross-sectional area had different influences on the importance coefficients of different components and had the strongest influence on the importance coefficient of the IHC. The importance coefficient of the IHC increased from 0.21 to 0.24 (14%) when the cross-sectional area was increased from 0.5 to 1.5 times the initial value. Thus, the cross-sectional area of a Cable-Strut element had a marginal effect on the collapse resistance and on the importance coefficients of components.

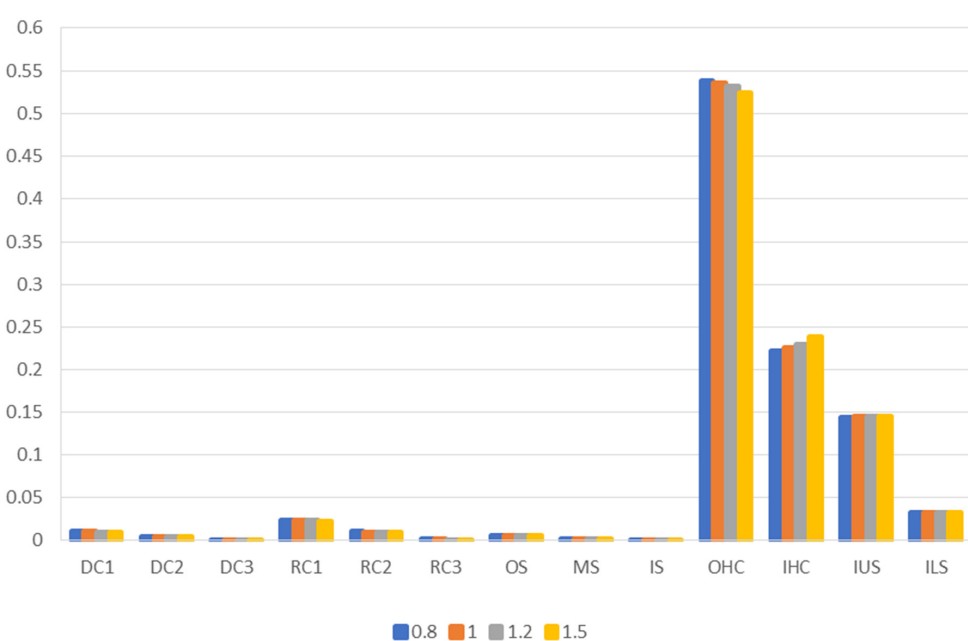

**Figure 7.** Importance coefficients of different elements under different cross-sectional areas of the elements.

### 4.2. Influence of the Radius of the Hoop Cable on the Collapse Resistance

The effects of the radius of the IHC on the importance coefficients of the Cable-Strut elements were first examined, and the other parameters were fixed in this examination. Figure 8 depicted the variations in the importance coefficients of the Cable-Strut elements when the radius of the IHC was varied between 0.8 and 1.2 times its initial value. The radius of the IHC had different influences on the importance coefficients of different components and had the strongest effect on the importance coefficient of the IHC. The importance coefficient of the IHC increased from 0.22 to 0.25 (14%), and that of the OHC decreased from 0.55 to 0.51 (7%) when the radius of the IHC was increased from 0.8 to 1.2 times its initial value. The variations in the importance coefficients of the other components were less than 7%. The aforementioned results indicated that the radius of the IHC had a weak influence on the collapse resistance of the investigated structure. Similarly, the authors found that the radius of the OHC also had a weak influence on the collapse resistance of the investigated structure.

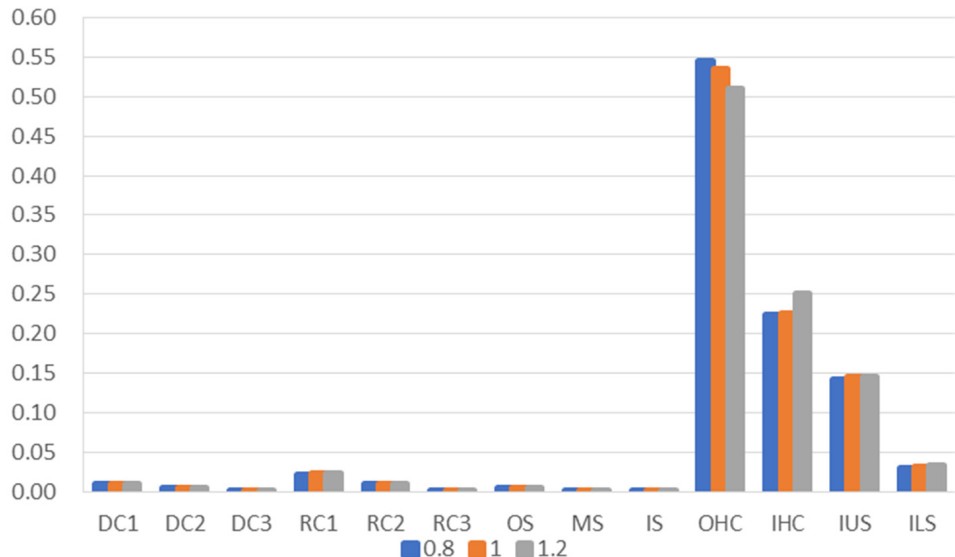

**Figure 8.** Importance coefficients of different elements under different radius of the inner hoop cable (IHC).

### 4.3. Influence of Strut Length on the Collapse Resistance

The influences of the length of the MS on the importance coefficients of the Cable-Strut elements were first examined, and the length of the MS was adjusted by changing the coordinates of its lower node. Figure 9 illustrates the variations in the importance coefficients of each Cable-Strut element when the length of the MS was varied to be between 0.8 and 1.2 times its initial value (the initial prestress of the OHC was fixed). As the length of the MS was increased from 0.8 to 1.2 times its initial value, the importance coefficient of the middle diagonal cable (DC2) increased from 0.0056 to 0.01 (79%), the importance coefficient of the outer ridge cable (RC1) decreased from 0.024 to 0.013 (54%) and the importance coefficients of the IHC and OHC varied by less than 7%. The aforementioned results indicated that the length of the MS had strong influences on the importance coefficients of some components but had a weak influence on the collapse resistance of the entire investigated structure. Similarly, the length of the OS also had a weak influence on the resistance of the investigated structure to progressive collapse resistance.

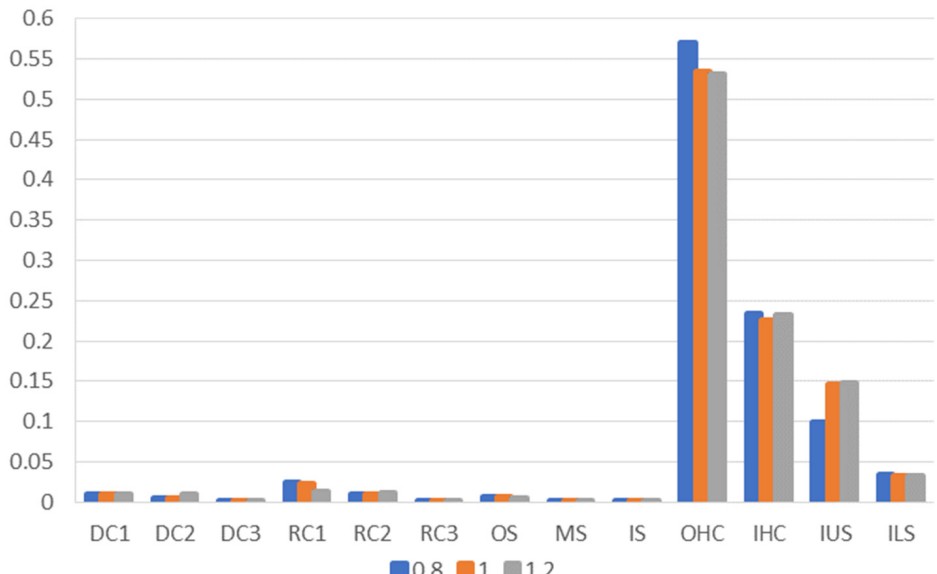

**Figure 9.** Variations in the importance coefficients of different elements with variations in the length of the MS.

### 4.4. Influence of Number of Structural Trusses on the Collapse Resistance

The influences of the number of trusses (between 12 and 24) on the importance coefficients of all the Cable-Strut elements were examined (Figure 10). The number of trusses weakly and strongly affected the importance coefficients of elements with high and low importance coefficients, respectively. As the number of trusses was increased from 12 to 24, the importance coefficients of the outer diagonal cable (DC1) and outer ridge cable (RC1) increased from 0.0067 to 0.053 (691%) and from 0.019 to 0.046 (142%), respectively; however, the importance coefficients of the OHC, IHC, and the IUS only varied between 10% and 21%. The collapse mode remained the same, but the collapse area varied when the number of structural trusses was changed. When the number of trusses was 24, the collapse area caused by the removal of the outer diagonal cable (DC1) and outer ridge cable (RC1) was 8.3%. However, when the number of structural trusses was reduced to 12, the collapse area increased to 16.6%. Thus, that outer diagonal cable (DC1) and outer ridge cable (RC1) transformed from common components into important components when the number of trusses was decreased from 24 to 12. The aforementioned results indicated that the number of structural trusses remarkably affected the importance coefficients of Cable-Strut elements and the resistance of the investigated structure to progressive collapse.

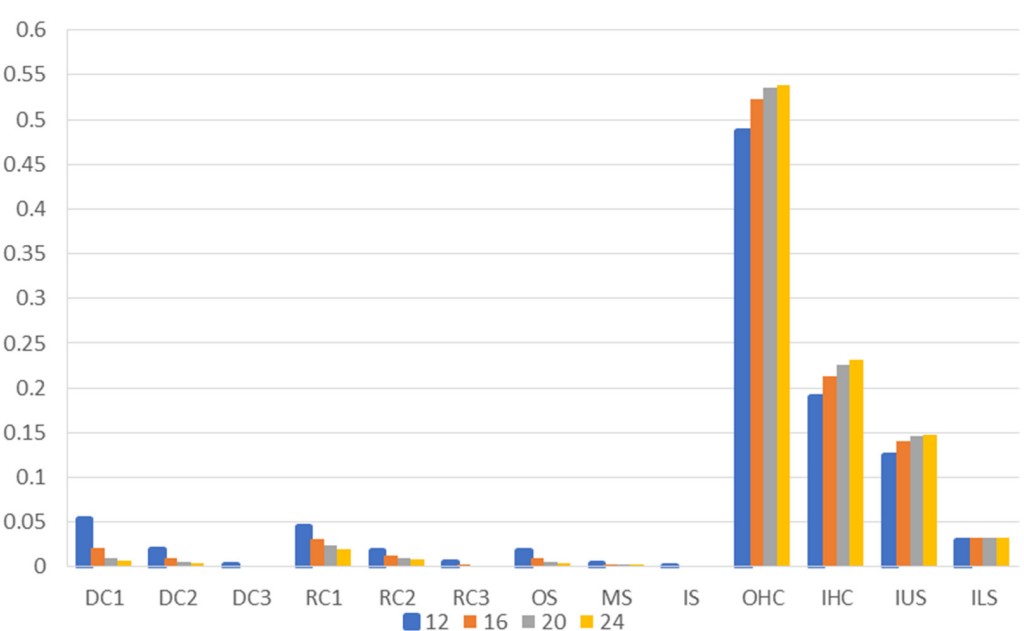

**Figure 10.** Variations in the importance coefficients of the elements with the number of structural trusses.

### 4.5. Influence of Structural Topological Relations on Collapse Resistance

To study the influence of topological relations on collapse resistance, four dome models with different topological relations were developed for a simulation: Dome 1, which represented a Geiger dome in Section 2; Dome 2, which represented a dome in which all the Cable-Strut elements surrounding the OHC were Levy-type elements; Dome 3, which represented a dome in which all the Cable-Strut elements surrounding the IHC were Levy-type elements; and Dome 4, which represented a dome in which all the elements were Levy-type elements (Figure 11). The other parameters of the structure, including the initial prestress of the outer loop, were kept constant in these models. The importance coefficient of each Cable-Strut element of the aforementioned models is depicted in Figure 12, and the results indicated the following. (1) Considerable variations were noted in the importance coefficients for the different dome models. The importance coefficients of the diagonal cable, ridge cables, hoop cables and tension hoop varied considerably when the topological relationship of the dome structure changed from a Geiger-type relationship to a Levy-type relationship. (2) The collapse areas and collapse modes of the four dome models differed considerably. For example, when the OHC was removed, the collapse area of Dome 2 decreased from 100% to 65.9%, and the maximum displacement also decreased. When the IHC was removed, the collapse area remained the same, and the maximum displacement marginally decreased. Consequently, the importance of the OHC decreased by 31%, whereas that of the IHC increased by 57%. The importance coefficients of the OHC, the IUS, the IHC and the inner lower string (ILS) for Dome 3 were 9% lower, 27% lower, 26% higher and 100% higher, respectively, than the corresponding values for Dome 1. In addition, the collapse areas formed because of the removal of the OHC and IHC were 100% and 47%, respectively, for Dome 1, and 62.5% and 42.3%, respectively, for Dome 3. (3) The elements of Dome 4 had similar importance to the elements of Dome 3. The main difference was that the redundancy of the IHC improved after the IHC was made to have a Levy-type topology, which resulted in the weakening of the influence of the IHC on the structure. The importance coefficients of the IHC and OHC for Dome 4 were marginally higher and those of the IUS and ILS for Dome 4 were marginally lower than the corresponding values obtained for Dome 3. In general, the topological relationship of a dome structure greatly affected the collapse mode and the importance coefficients of structural elements.

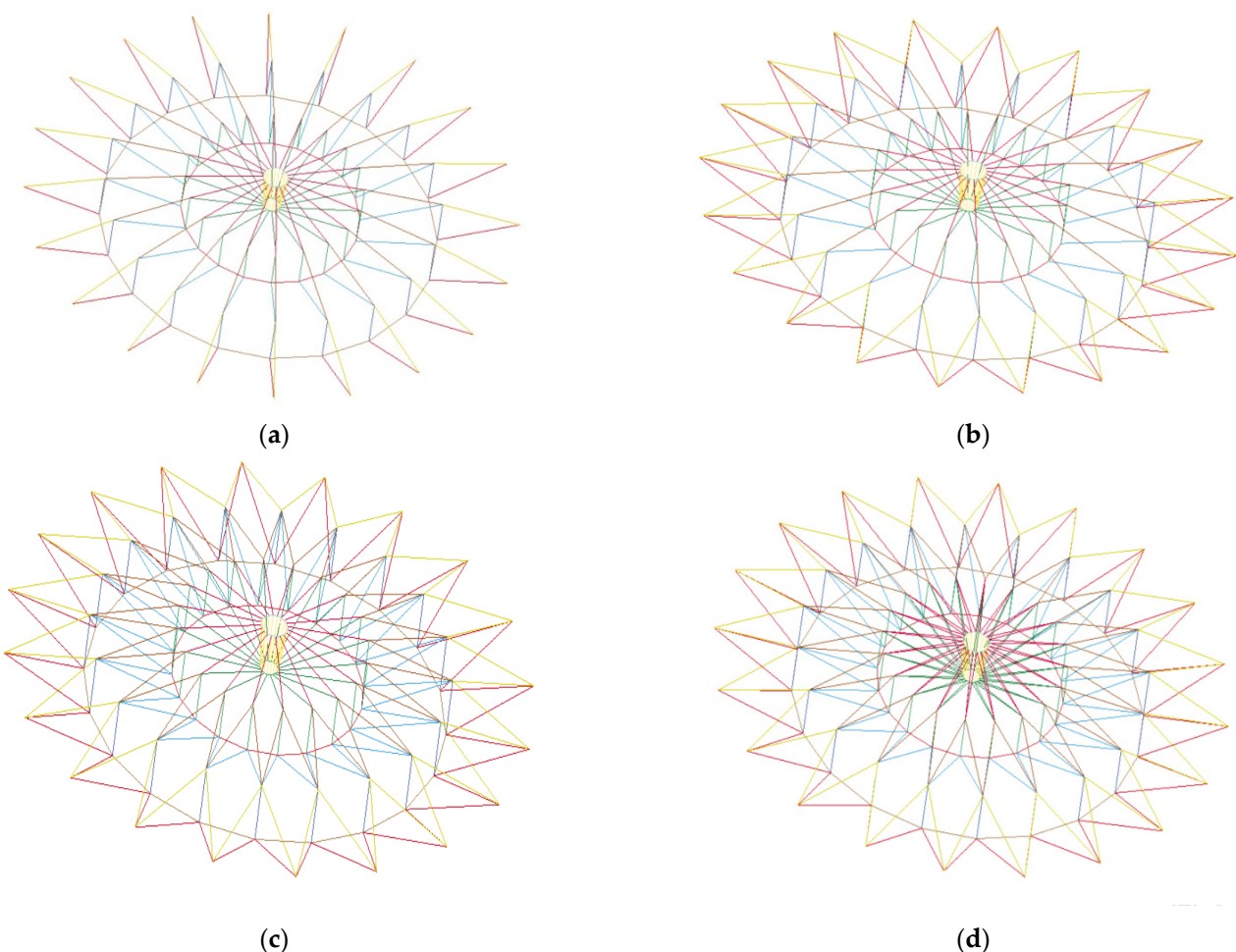

**Figure 11.** Topological relation diagram for the four developed cable dome models. (**a**) Dome 1, (**b**) Dome 2, (**c**) Dome 3, (**d**) Dome 4.

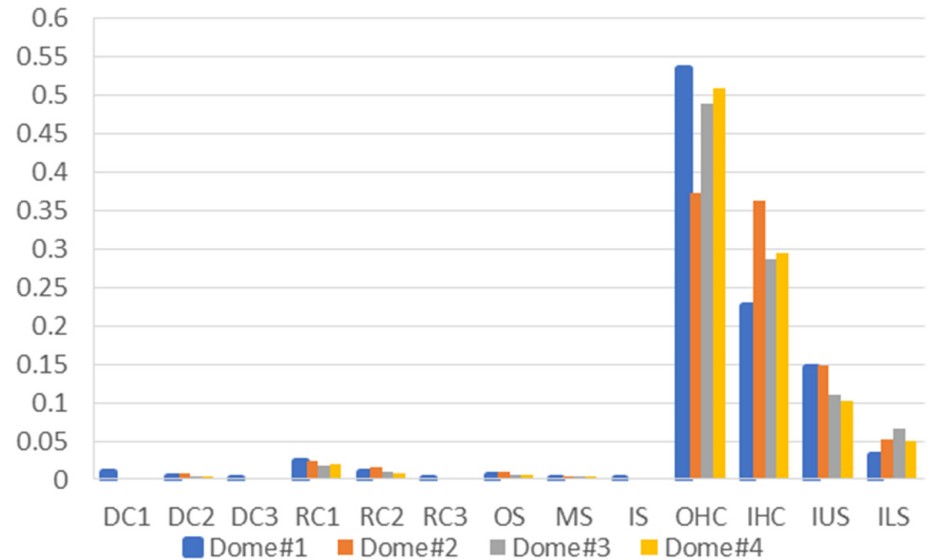

**Figure 12.** Variations in the importance coefficients of the Cable-Strut elements under different topological relationships for a dome structure.

## 5. Critical Value of the Element Importance Category

The aforementioned results indicated that the most important factors affecting the collapse mode of a cable dome and the importance coefficients of its elements were the number of trusses and the topological relationship. Parameters such as the cross-sectional areas of the Cable-Strut elements, the lengths of the struts and the radii of the hoop cables had a weaker influence on the collapse mode than the aforementioned parameters. To comprehensively understand the factors affecting element importance, this study analysed the influences of the structure type and topological relation on the critical value of the element importance category.

### 5.1. Critical Value of the Importance Coefficient for Dome 1

The structural dynamic response and collapse mode exhibited by Dome 1 when the Cable-Strut elements were removed (Table 2) indicated that the OHC and IHC were key components, the IUS was an important component, and the remaining Cable-Strut elements were common components. According to the results displayed in Figure 13, the critical importance values for distinguishing between key and important components and between important and common components for Dome 1 were 0.18 and 0.09, respectively. Thus, importance coefficients >0.18, between 0.09 and 0.18, and <0.09 corresponded to key, important and common elements for Dome 1, respectively.

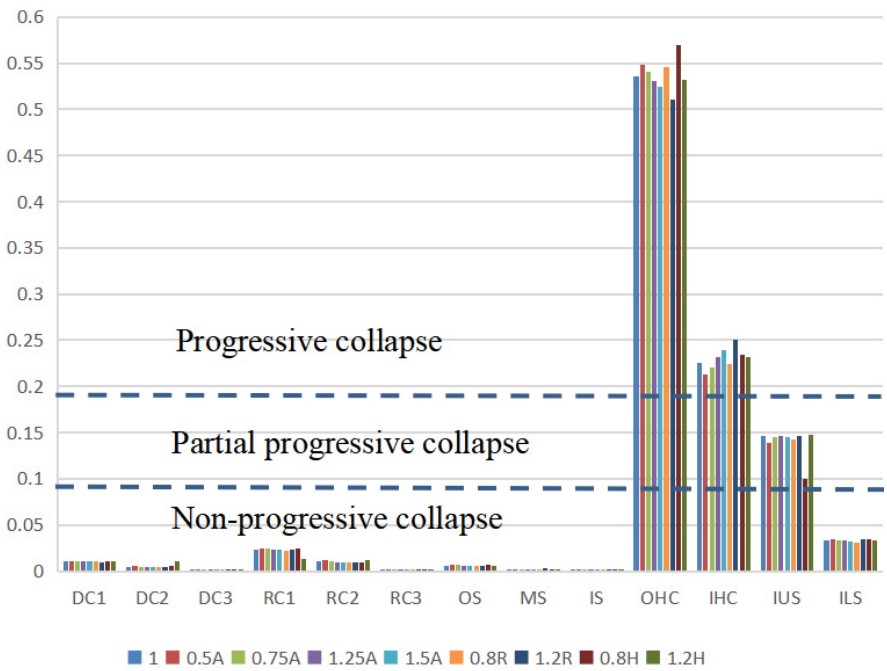

**Figure 13.** Importance coefficients of the structural elements under different cross-sectional areas of cables, radii of the IHC and lengths of the MS.

### 5.2. Influences of the Number of Structural Trusses and Topological Relations

Table 3 presents the simulation results on the collapse modes and collapse areas under different numbers of trusses and different topological relationships. (1) When the number of structural trusses was reduced from 24 to 12, the collapse area formed because of the removal of the outer ridge cable (RC1) or outer diagonal cable (DC1) increased from 8.3% to 16.6% (an increase of more than 15%), and the classification of these elements changed from common components to important components. The collapse areas formed because of the removal of the remaining elements had no effect on the importance categories of the structural elements. (2) Moreover, when the topological relationship was adjusted from Dome 1 to Dome 3 or Dome 4, the collapse area caused by the removal of the IUS decreased from 17.4% to 13.1% or 0%, respectively (both less than 15%). Thus, the aforementioned

change in the topological relationship resulted in a change in the classification of the IUS from an important component to a common component. The collapse areas caused by the removal of the other elements did not affect the importance categories of the structural elements.

**Table 3.** Collapse areas formed because of the removal of different structural elements under different topological relationships.

| Topologies Element | Dome 1 with 12 Strusses | Dome 1 with 16 Strusses | Dome 1 with 20 Strusses | Dome 1 with 24 Strusses | Dome 2 | Dome 3 | Dome 4 |
|---|---|---|---|---|---|---|---|
| DC1 | 16.60% | 12.50% | 10% | 8.30% | 0% | 0 | 0 |
| DC2 | 0 | 0 | 0 | 0 | 0 | 0 | 0 |
| DC3 | 0 | 0 | 0 | 0 | 0 | 0 | 0 |
| RC1 | 16.60% | 12.50% | 10% | 8.30% | 10% | 0 | 0 |
| RC2 | 7.70% | 5.80% | 4.60% | 3.8% | 4.60% | 4.60% | 0 |
| RC3 | 0 | 0 | 0 | 0 | 0 | 0 | 0 |
| OS | 14.4% | 10.9% | 0 | 0 | 0 | 0 | 0 |
| MS | 0 | 0 | 0 | 0 | 0 | 0 | 0 |
| IS | 0 | 0 | 0 | 0 | 0 | 0 | 0 |
| OHC | 100% | 100% | 100% | 100% | 65.9% | 62.5% | 62.5% |
| IHC | 47% | 47% | 47% | 47% | 47% | 42.3% | 42.3% |
| IUS | 20.3% | 18.5% | 17.4% | 16.7% | 17.4% | 13.1% | 0 |
| ILS | 0 | 0 | 0 | 0 | 0 | 0 | 0 |

The importance coefficients and importance categories of the Cable-Strut elements were examined under three structural topologies: Dome 1 with 12 trusses, Dome 3 and Dome 4 (Table 4). When the number of trusses in Dome 1 was optimised from 20 to 12, the critical value separating the key components and important components became 0.16 and that separating the important components and common components became 0.04. Domes 3 and 4 had no important components, and the critical value separating the key components and common components was 0.2. The aforementioned results indicate that the number of structural trusses and the topological relation strongly influence the critical value separating the importance categories of Cable-Strut elements.

**Table 4.** Importance coefficients and importance categories of the structural elements for Dome 1 with 12 trusses, Dome 3 and Dome 4.

| Element | Dome 1 with 12 Trusses | | Dome 3 | | Dome 4 | |
|---|---|---|---|---|---|---|
| | Importance Coefficients | Component Properties | Importance Coefficients | Component Properties | Importance Coefficients | Component Properties |
| DC1 | 0.053 | Important component | 0.002 | Common component | 0.0021 | Common component |
| DC2 | 0.02 | Common component | 0.0037 | Common component | 0.0036 | Common component |
| DC3 | 0.0032 | Common component | 0.0018 | Common component | 0.00079 | Common component |
| RC1 | 0.046 | Important component | 0.019 | Common component | 0.019 | Common component |
| RC2 | 0.018 | Common component | 0.0095 | Common component | 0.0088 | Common component |
| RC3 | 0.005 | Common component | 0.0029 | Common component | 0.00081 | Common component |
| OS | 0.018 | Common component | 0.0066 | Common component | 0.0069 | Common component |
| MS | 0.004 | Common component | 0.0035 | Common component | 0.0033 | Common component |
| IS | 0.00026 | Common component | 0.00017 | Common component | 0.00014 | Common component |
| OHC | 0.49 | Key component | 0.49 | Key component | 0.51 | Key component |
| IHC | 0.19 | Key component | 0.29 | Key component | 0.29 | Key component |
| IUS | 0.13 | Important component | 0.11 | Common component | 0.1 | Common component |
| ILS | 0.03 | Common component | 0.066 | Common component | 0.049 | Common component |

## 6. Conclusions

In this study, the collapse mechanisms of cable domes and the importance of various Cable-Strut elements in the development of structural resistance to progressive collapse were examined. The main results of this study were as follows.

(1) Different dynamic responses and collapse modes were induced by the removals of different types of Cable-Strut elements. The strongest dynamic responses were induced by the removal of the IHC and OHC, followed by the removal of the IUS. The dynamic responses induced by the removal of other types of Cable-Strut elements were relatively weak. Thus, different Cable-Strut elements differed in their importance to the resistance of cable dome structures to progressive collapse.

(2) Cable-Strut elements differed in their importance coefficients. For the investigated cable dome structure, the OHC and IHC had the highest importance coefficients. This structure underwent progressive collapse when the aforementioned elements were removed; thus, the OHC and IHC were key components. The importance coefficient of the IUS was smaller than those of the aforementioned elements. Nevertheless, local collapse occurred when the IUS was removed; thus, this element was an important element. Other types of Cable-Strut elements had considerably smaller importance coefficients than did the aforementioned three elements; thus, their removals did not cause progressive collapse. Consequently, these elements were common elements.

(3) Different cable dome structures had different critical values of the element importance category. For the investigated Geiger dome (Dome 1), the critical values for distinguishing between key components and important components and between important components and common components were 0.18 and 0.09, respectively.

(4) A structure's resistance to progressive collapse, the importance categories of Cable-Strut elements and the critical values separating importance categories were weakly influenced by structural design parameters, such as the cross-sectional area of an element, the length of a strut and the radius of a hoop cable, but were strongly influenced by the number of trusses and the topological relations of the structure. Thus, a conservative approach should be adopted when selecting the number of trusses and designing the topological relations for a structure.

**Author Contributions:** Conceptualization, L.C.; methodology, L.C. and Y.L.; software, Z.L. and K.H.; validation, Y.L., K.H. and Y.Z. (Yihong Zeng); resources, L.C., Y.Z. (Yiyi Zhou) and S.D.; writing—original draft preparation, L.C. and Y.L.; writing—review and editing, L.C. and Z.L.; supervision, L.C.,Y.Z. (Yiyi Zhou) and S.D.; project administration, L.C. All authors have read and agreed to the published version of the manuscript.

**Funding:** This study was supported by the National Natural Science Foundation of China (Grant No. 51578422, 51678082).

**Institutional Review Board Statement:** Not applicable.

**Informed Consent Statement:** Not applicable.

**Data Availability Statement:** Not applicable.

**Conflicts of Interest:** The authors declare no conflict of interest.

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
