# Peer review of "Analysis and Evaluation of the Progressive Collapse Behaviour of a Cable Dome Structure"

_buildings, doi:10.3390/buildings12101700_

Round 1

Reviewer 1 Report

The manuscript reports a numerical study on progressive collapse behavior of a cable dome structure. Dynamic responses and collapse modes under different member removal cases are reported and discussed. A framework for element importance classification based on collapse modes is developed. The topic is very interesting, since the studies focusing on the progressive collapse of non-building structures are quite limited. The manuscript is well-written in general. However, the following points should be carefully addressed before any consideration for publication.

1.     In L29, Fuller's idea of tensegrity needs a reference. Also, for Geiger cable in L98.

2.     Please pay more attention to the size of the paragraphs. Just as example, in the Section 1 a paragraph started in L36 and ended in L96. Such details can affect the readability of your manuscript.

3.     In L100 please clarify which code are used in the numerical study, LS-DYNA or Ansys? In general, they are two different packages.

4. Literature review is limited and can be improved. A recent comprehensive review on the topic is reported in [A]. Research works on progressive collapse of tensegrity systems are available in [B, C]. Please enrich the review accordingly.

[A] Kiakojouri, F., De Biagi, V., Chiaia, B., & Sheidaii, M. R. (2022). Strengthening and retrofitting techniques to mitigate progressive collapse: A critical review and future research agenda. Engineering Structures, 262, 114274.

[B] Shekastehband, B., Abedi, K., Dianat, N., & Chenaghlou, M. R. (2012). Experimental and numerical studies on the collapse behavior of tensegrity systems considering cable rupture and strut collapse with snap-through. International Journal of Non-Linear Mechanics, 47(7), 751-768.

[C] Shekastehband, B., Abedi, K., & Chenaghlou, M. R. (2011). Sensitivity analysis of tensegrity systems due to member loss. Journal of Constructional Steel Research, 67(9), 1325-1340.

5.     Please provide brief background regarding the primary design of the model structure (the sports center). Which parameters and loads are considered in the design phase? For example, is any seismic requirement included in the design? Such details can help to understand structural behavior under progressive collapse scenarios more deeply.

6.     Member removal cases should be provided in a table or a figure.

7.     Converting the Table 2 to an equivalent figure can be more illustrative.

8.     Shorter removal time usually leads to more dynamic effects and consequently larger displacement. Why is the period rounded to 2 instead of 1 second? The later leads to a more critical scenario.

9.     Since LS-DYNA solely equipped with dynamic explicit solver, the application of gravity (and other static loads) can be tricky. How do the authors solve this issue?

10.  No verification and validation are provided in the manuscript. In this situation the accuracy of the numerical techniques and reported results is under question.

Reviewer 2 Report

This paper is very interesting and important of civil engineering. In my opinion this topic is very important to behavior of geodesic dome, thus please compare your research to the geodesic dome (lightweight structure) of their dynamic behavior. Below you can find paper about lightweight geodesic dome under dynamic loads:

http://doi.org/10.5120/ijca2016909346

https://doi.org/10.1142/S021945541440001X

https://doi.org/10.3390/ma14164493

Finally, I hope that my comments were helpful to Authors.

Round 2

Reviewer 1 Report

The manuscript is sufficiently improved to warrant publication.

Reviewer 2 Report

Thank you for your improving.